# Swarm Intelligence in Internet of Medical Things: A Review

**DOI:** 10.3390/s23031466

**Published:** 2023-01-28

**Authors:** Roohallah Alizadehsani, Mohamad Roshanzamir, Navid Hoseini Izadi, Raffaele Gravina, H. M. Dipu Kabir, Darius Nahavandi, Hamid Alinejad-Rokny, Abbas Khosravi, U. Rajendra Acharya, Saeid Nahavandi, Giancarlo Fortino

**Affiliations:** 1Institute for Intelligent Systems Research and Innovation, Deakin University, Waurn Ponds, Geelong, VIC 3216, Australia; 2Department of Computer Engineering, Faculty of Engineering, Fasa University, Vali asr Blvd, Fasa 74617-81189, Iran; 3Department of Electrical and Computer Engineering, Isfahan University of Technology, Daneshgah e Sanati Hwy, Isfahan 84156-83111, Iran; 4Department of Informatics, Modeling, Electronics and Systems (DIMES), University of Calabria, 87036 Cosenza, Italy; 5BioMedical Machine Learning Lab (BML), The Graduate School of Biomedical Engineering, UNSW, Sydney, NSW 2052, Australia; 6UNSW Data Science Hub, The University of New South Wales (UNSW Sydney), Sydney, NSW 2052, Australia; 7Health Data Analytics Program, AI-Enabled Processes (AIP) Research Centre, Macquarie University, Sydney, NSW 2109, Australia; 8Department of Electronics and Computer Engineering, Ngee Ann Polytechnic, Singapore 599489, Singapore; 9Department of Biomedical Engineering, School of Science and Technology, SUSS University, Singapore 599494, Singapore; 10Department of Biomedical Informatics and Medical Engineering, Asia University, Taichung 41354, Taiwan; 11Harvard Paulson School of Engineering and Applied Sciences, Harvard University, Allston, MA 02134, USA

**Keywords:** internet of things, internet of medical things, internet of things in health, swarm intelligence algorithm, wearable devices, wireless sensor network

## Abstract

Continuous advancements of technologies such as machine-to-machine interactions and big data analysis have led to the internet of things (IoT) making information sharing and smart decision-making possible using everyday devices. On the other hand, swarm intelligence (SI) algorithms seek to establish constructive interaction among agents regardless of their intelligence level. In SI algorithms, multiple individuals run simultaneously and possibly in a cooperative manner to address complex nonlinear problems. In this paper, the application of SI algorithms in IoT is investigated with a special focus on the internet of medical things (IoMT). The role of wearable devices in IoMT is briefly reviewed. Existing works on applications of SI in addressing IoMT problems are discussed. Possible problems include disease prediction, data encryption, missing values prediction, resource allocation, network routing, and hardware failure management. Finally, research perspectives and future trends are outlined.

## 1. Introduction

In the modern era, addressing the healthcare needs of the ever-growing global population is challenging in terms of cost and medical asset accessibility. Establishing the infrastructure for remote healthcare services is one of the major factors in dealing with global healthcare challenges. Leveraging existing technologies such as internet of things (IoT) is a natural choice to accelerate the deployment of remote healthcare services [1]. IoT is based on technologies such as cloud computing, wireless sensor networks (WSNs), radio-frequency-identification (RFID) devices [2], etc., providing information exchange, data processing/storage, and decision-making. However, IoT development faces serious challenges such as security and energy consumption due to the large number of IoT devices. Considering the complexity and dynamic nature of the IoT, swarm intelligence (SI) algorithms may be used to address its challenges [3]. SI algorithms have been inspired by organized and intelligent behaviors in nature that groups of birds, ants, bees, etc., exhibit. Although each member in the group has limited ability, the entire group is able to perform complex tasks via cooperation and information exchange [4]. This type of behavior is called swarm intelligence. One of the prominent features of SI algorithms is their parallelism and distributed characteristics, making them suitable for IoT applications [5]. In the IoT setup, several devices with possibly limited processing capabilities are present. Exploiting the swarm intelligence can ease the implementation of distributed data processing across resource-constrained devices, leading to efficient resource utilization. SI methods can be used for optimization requried by the growing development of the IoT [6]. Using IoT in medical domain leads to internet of medical things (IoMT). In this article, the key research related to applications of SI algorithms in the IoT domain with a special focus on IoMT is reviewed. Given that WSN is one of the primary technologies for IoT realization, the SI applications in four major subcategories of WSN, i.e., node localization, sensor deployment, routing protocol, and CH selection, are reviewed. Appropriate coverage of the environment is an important factor in WSNs deployment. However, the number of static nodes is limited and may not be enough to cover the whole area appropriately. The existing literature on the dynamic coverage of the environment using a swarm of unmanned aerial vehicles (UAVs) [7] is reviewed as well. Apart from dynamic coverage, other issues related to WSNs such as energy limitation are also reviewed.

The ultimate goal of IoMT is providing better medical services to patients. One of the critical requirements of IoMT is that it must be affordable and available anywhere and at anytime. Such requirements can be realized by exploiting portable devices such as wearables that IoMT users already own. The applications of wearables in IoMT are reviewed as well. The article is ended with closing thoughts on possible future directions for further development of IoMT.

### 1.1. Search Strategy

We conducted our search by relying on the Scopus search engine. The statistics of the conducted search have been shown in Figure 1, which has two rows of numbers. The top row of numbers represents the number of candidates’ (selected) papers, and the bottom row is the sum of candidate papers and papers that have been selected from the references of the candidate papers. Each paper was studied by four authors. The papers confirmed by at least three authors were included in this review. The paper selection was done based on criteria such as paper publisher, citation count, and relevance to IoT/IoMT/SI.

As mentioned before, only a subset of the papers collected during our search has been included in this review. Nevertheless, reporting the statistics over all of the collected papers related to the application of SI in IoT reveals interesting information about the percentage of papers published in different research fields (Figure 2a) and their document types (Figure 2b). The number of papers published per year is also available in (Figure 2c). As seen in Figure 2a, SI+IoT is mostly used in computer science and engineering, which shows that SI methods are better suited to optimization problems in these two fields. Careful inspection of Figure 2b reveals that the number of review papers devoted to applying SI in IoT is limited. This shows that much work must be done to aggregate and summarize existing research in the SI+IoT field. In Figure 2c, the growing trend of the number of papers published per year is evident.

Paper statistics related to the application of SI in IoMT have also been presented in Figure 3. As seen in Figure 3b, the number of papers on this field is limited, which shows that IoMT is still in the early stages of development and that much work needs to be done in this field.

### 1.2. Contributions

Many studies have outlined relevant developments in SI-oriented applications but have only considered a small number of algorithms [8,9]. Some of the most important SI algorithms, such as routing algorithms based on particle-swarm optimization (PSO), are absent from existing reviews. The survey of Muhammad et al. [10] was exclusively devoted to routing protocols in WSN. Their main contribution was a general recipe for the scientific definition of sound experiments and performance measures, and the fact that SI applications in WSNs will grow in the future. However, the authors did not discuss sensor-location and CH-selection issues. Sandra et al. [11] reviewed applications of various bio-inspired algorithms in WSNs. The main contribution was the careful inspection of routing, fault tolerance, node localization, data collection, etc., based on SI methods. The authors have even provided the percentage that each SI method has used when addressing different issues in WSNs. Zedadra et al. [12] took a step further and investigated the application of SI methods in IoT. They categorized existing SI methods into three groups. The first group is the set of methods, e.g., ant-colony optimization (ACO) [13], PSO [14], and artificial bee colony (ABC) [15], with wide application in the IoT domain. The second group is the set of methods that have high application potential but have been used rarely in practice and deserve more attention. The third group consists of SI methods that have not reached acceptable maturity yet. Zedadra et al. suggested that these methods need further development before they can be used in real-world problems.

Rahouma et al. [16] investigated the challenges of using the social internet of things (SIoT) in the healthcare domain. They introduced two new applications in the field of IoMT. The first application is related to heart disease diagnosis and the second to brain tumors diagnosis. Based on the traveling sales problem, ACO has been used to extract features for binary MRI images of various brain tumor cases. The results of the two applications prove the importance of the proposed methods in IoMT. The review papers mentioned in this section and some additional works have been summarized in Table 1 to facilitate the comparison of this survey with existing ones. Based on the data provided in Table 1, the major difference between our review and previous ones is that we investigate the application of SI in IoT with a special focus on the IoMT domain.

The rest of the paper is structured as follows: an overview of IoT is presented in Section 2. In addition, IoMT and the prerequisites of IoT are described. The SI algorithms are reviewed in Section 3. Various SI algorithms applied to IoT/IoMT problems are reviewed in Section 4. The current situation and development trends are discussed in Section 5. The paper ends with the conclusion in Section 6.

## 2. IoT Overview

The IoT consists of four primary components, which are perception, connectivity, data processing, and user interface as shown in Figure 4. In the perception component, various sensors and devices such as GPS, QR-Code, RFID, etc., are used to gather vital information from the environment in which IoT operates. The transfer of collected data to users and/or cloud servers for further processing and storage is performed using communication technologies such as Bluetooth, Zigbee, and WLAN. The collected data are processed using a combination of cloud and edge computing. The IoT interface with users is realized via different devices such as smartphones, smartwatches, microphones, cameras, monitors, etc. [21].

### 2.1. IoT Prerequisite

Before diving into the studies related to IoT and SI, it is beneficial to provide an overview of IoT prerequisites.

#### 2.1.1. Wireless Sensor Network

WSN is a wireless network consisting of many self-organizing and multi-hop static or mobile sensors. This network collects, processes, and transmits information to users. Due to features such as low cost and high compatibility, WSN has gained popularity in recent years. Existing applications of WSN include environmental monitoring, agricultural production, military, medicine, and industry. In WSN, each sensing device has a limited battery capacity [22], so it is crucial to use an efficient routing protocol, cluster-head selection, and node localization. To this end, the SI algorithms’ ability to find optimal paths inspired by swarm living organisms can be exploited. These algorithms have the robustness and scalability required by the WSN routing protocol. Given that SI is capable of finding local optima for NP-hard problems, it can tackle the NP-hard problem of cluster-head (CH) selection in WSN. Additionally, SI multi-objective optimization capability can find CH in multiple clusters simultaneously [23].

#### 2.1.2. WSN Routing Protocols

Unlike conventional wireless networks, the WSNs are often large-scale, and their nodes have random locations, low computing and communication capabilities, and limited energy. Therefore, routing protocols must be designed efficiently to save energy by considering the communication path between the command center and the sensor nodes. In addition, protocols need to be self-organized and fault-tolerant. SI algorithms have such properties and can address WSN complex nonlinear problems. WSN routing protocols are either hierarchical or planar [24]. In a hierarchical approach, networks are divided into clusters with variable sizes. Each cluster contains one CH and several members. The head of the cluster manages other member nodes. The sensor nodes of each cluster send perceived data to the CH to be transferred to a sink node or BS. The most common SI algorithms used For CH selection are PSO, ACO, and ABC.

### 2.2. IoMT Overview

In the modern era, the life expectancy of the global population has increased, but people usually live longer with comorbidities needing continuous medical treatments. The cost of these treatments is increasing worldwide, which urges developing efficient and economic approaches to provide people with appropriate medical treatments. The advancements in wireless technology, processing power, and device miniaturization have paved the road toward developing connected (possibly portable) medical devices capable of data collection, analysis, and transmission to servers of healthcare centers. As shown in Figure 5, combining these devices, medical software, and connectivity technologies has given birth to the internet of medical things (IoMT) [25]. The first stage of IoMT is data acquisition, in which medical information is collected from patients via devices such as blood-pressure sensors, accelerometers, and body-temperature sensors. The second stage involves data pre-processing. For example, sensor readings may be analog and should be converted into digital form. Additionally, sensor readings are usually accompanied with noise which must be filtered. In the next stage, data will be sent to data centers for storage, where medical experts can access them for further analysis. Thanks to automated data pre-processing and aggregation, the medical experts will have a much easier job analyzing a large volume of medical data belonging to many patients [26].

Part of the IoMT is a set of devices with various functionalities [27]. For example, smart thermostats can be used to keep the room temperature at an optimum level. Smart light bulbs will turn on and off as needed leading to reduced electricity consumption. The maintenance of healthcare centers may also be considered as part of IoMT. Using a 3D model of the healthcare center makes it possible to quickly locate and fix malfunctions in the center infrastructure. The majority of patients’ interaction with healthcare systems is via devices developed by MedTech companies. Examples of these devices are wearables, implantables, and stationary devices. Wearables are devices that can be worn by their users. The primary objective behind using wearables is having access to specific services no matter where or when. Wearables may serve a variety of purposes in the IoMT. For example, real-time location services reveal the patient’s location, which is very helpful in case the patient’s condition becomes critical and immediate medical attention is required [28]. The patient’s condition can be determined based on his/her vital signs monitored by appropriate sensors. As mentioned before, taking a precise dose of medicine is critical for some diseases such as diabetes. Smart-drug-delivery devices can help with that matter. For example, patients with Asthma can benefit from connected inhalers. These inhalers are connected to special purpose applications on patients’ smartphones and are designed to measure lung function and disease progression [29].

IoMT and wearables are not only active fields of research but are massively impacting daily life and economic sectors. IoMT global market share is a growing trend. The market share in the Asia-Pacific region is the largest; it grew from $11 billion in 2017 to $51 billion in 2022 [30]. The runner-up is North America, with an expected market share of $45 billion in 2022. Europe claims third place with a $44 billion market share. Less developed countries in South America, the Middle East, and Africa have the smallest market share.

#### Benefits

IoMT has huge potential for revolutionizing the healthcare systems [31] adopting medical technologies at large scales. IoMT can impact the traditional healthcare systems in multiple ways [32]. For example, patients with chronic diseases can be monitored by clinicians remotely without the need to stay in a hospital. Remote monitoring makes disease management easier for both patients and clinicians in terms of comfort and treatment costs. IoMT can ease the workflow of healthcare centers by facilitating data collection and processing, leading to an enhanced patient experience. Taking the appropriate dosage of medicine is critical to effectively controlling certain diseases. With the constant monitoring ability of IoMT, patients are not concerned about following the appropriate medicine regimens. Patients will always have access to the appropriate dose of their medicine, thanks to the constant monitoring of their conditions. The benefits of IoMT can be summarized as [30]:Improved drug managementDecreased treatment costEnhanced patient experienceImproved patient outcomeImproved diagnosis and treatmentRemote monitoring of chronic diseases

## 3. SI Algorithms

SI belongs to the family of meta-heuristic algorithms [33] inspired by the behavior of social species such as birds, termites, ants, and bees. SI is a complex, coordinated, and flexible collective behavior realized by a large group of agents that follow simple rules. The SI has been designed to achieve challenging objectives in a given task environment by relying on the synergy between simple agents. Although agents take local actions, their interaction with each other and the environment leads to achieving the final objective of the task environment. The SI workflow consists of exploitation and exploration phases [34]. During exploration, the agents search the environment by choosing random but goal-oriented actions leading to potentially better solutions in later optimization steps. During exploitation, the agents choose the best possible action given the skills they have learned so far. Figure 6 shows some of the most common and widely used SI algorithms, such as ACO [35], ABC [36], particle-swarm optimization (PSO) [37], etc.

Flexibility of SI algorithms let them cope with external challenges and internal disturbances. The swarm individuals may have different and possibly imperfect capabilities (e.g., in GNP, individuals have different decision-making flowcharts). Through synergy, the individuals will be able to cover each other’s weaknesses and reinforce each other’s strengths achieving challenging tasks. SI systems are scalable and may include a few to millions of individuals thanks to their decentralized decision-making. The colony can adapt to predefined stimuli as well as new ones. Heterogeneity is easily realized in SI systems because each individual has its own decision-making module. Therefore, having diverse individuals is only a matter of changing their decision-making module. Parallel execution is yet another feature of SI that is realized by the simultaneous decision-making of individuals [38].

Like any other AI approach, SI also has some drawbacks. For example, it is hard to predict the behavior from the rules of individuals. The goal of the swarm cannot be determined just by inspecting the individuals’ functionalities. In addition, a small change in a rule may result in drastically different behavior [39]. However, these drawbacks have not stopped SI algorithms from gaining popularity in different application domains since their proposal in the 1990s. ACO and PSO, which were introduced in 1992 and 1995, respectively, are two famous SI algorithms. From 2000 to 2010, multiple SI methods such as bacterial food optimization (BFO) [40], the artificial fish swarming algorithm [41], the firefly algorithm (FA) [42], and the ABC algorithm [43] have been proposed. Pigeon-inspired optimization (PIO) [44], the grey wolf optimizer (GW) [2], and the butterfly optimization algorithm (BOA) [45] were proposed in 2014, 2014, and 2015, respectively. In the following, the basic concepts of widely utilized SI optimizations, i.e., ACO, PSO, and ABC, are briefly explained.

PSO is inspired by the social search movements of several species, such as birds or fish. This method aims to achieve the optimal solution in a multidimensional search space. Initially, each particle in the population is assigned a random velocity and position. These particles gradually move towards (possibly) the global optimum through exploiting well-known positions and exploration [46].

ACO was inspired by ants’ communication mechanism via pheromones for finding a near-optimal path between the food source and the ant colony. This phenomenon is called ‘stigmergy’. The likelihood that ants would choose a specific route for reaching the food source is a function of the pheromone intensity and the distance between the food source and the colony. ACO utilizes the knowledge of previously traveled routes using pheromone traces on those paths and simultaneously explores new paths to develop candidate solutions [47].

The movement patterns of bees inspired the ABC algorithm development during food search. ABC divides the swarm of artificial bees into three types: scouts, onlookers, and employed bees. Each employed bee inspects food sources and decides on the nearest one, then evaluates the amount of nectar and informs the other bees of the quantity, quality, distance, and direction of the food source by dancing. Each onlooker bee observes the movement pattern of the employed bees, selects one of their sources based on the movement, and then travels to that source. The onlooker bee selects a nearby neighbor and estimates the amount of its nectar. Next, the scouts determine whether the onlooker is trapped in a local minimum in which case random search is conducted for another potential source of food. Each food source is a possible solution to the search problem, and the food source amount of nectar corresponds to its quality. The goal of ABC is to locate good food sources in a difficult search space [48].

## 4. Application of SI in IoT/IoMT

In this section, existing works on applications of SI methods in IoT/IoMT are reviewed. Some of the reviewed papers have utilized more than one SI algorithm. These papers are categorized according to their main algorithm. This section is structured based on the most popular SI methods, i.e., PSO, ACO, and ABC, applied in IoT/IoMT.

### 4.1. PSO in IoT/IoMT

The IoT consists of several modules with specific goals. Sensing the operational environment is critical for taking appropriate actions to achieve the system objectives. A wireless sensor network is a common approach widely used in IoT systems. Any network-oriented system needs efficient routing for data transfer between its nodes. In [49], a PSO-oriented WSN routing optimization algorithm has been proposed. By optimizing the number of nodes present in a cluster and the number of clusters formed, the network life is extended, and the energy consumption is reduced. The proposed algorithm minimized by the packets loss rate and the statistical end-to-end latency improve the node-survival rate and increase the number of clusters. Another PSO-oriented WSN routing method [50] combines mobile sink technology with virtual clustering in the routing. Three formats of packets are defined. One of these packets determines the cluster by which the data must be sent to the mobile sink. The other packet is the message sent directly by CH to the mobile sink, and the last packet contains the message sent to CH by the member node. The cluster with the highest residual energy is used for information collection for sink node. The member node transmits the information to CH when CH is determined by the sink.

Fault occurrence during packet routing is inevitable. Therefore, it is desirable to have a fault-tolerant routing method [51]. This routing method modifies PSO by improving the existing multi-objective SI algorithms with rapid recovery from the failure path. Sensors calculate objective function values, and multiple paths are constructed to optimize these values such that average delay and overall energy consumption are minimized. The main drawback of this method is that it needs a long time to establish the objective function at the beginning of the iteration.

Covering a specific area with a network of sensors is widely practiced in IoMT. Wang et al. [52] presented a PSO-oriented coverage method in static WSNs with randomly positioned sensors. Building a WSN for full coverage of a critical grid using the minimum number of sensors deployed on grid points is NP-complete [53]. In this method, networks are divided into grids, and the coverage rate of each grid is calculated. The node-sensing range is adjusted until the coverage rate reaches at least 90%. The radius of sensing is optimized and adjusted by the PSO algorithm. The results indicate that this algorithm can improve the area coverage and reduce energy consumption.

Efficient energy consumption is a critical design factor in WSNs. That is why Wang et al. [54] proposed a clustering method that employed PSO to prevent energy-hole generation and perform CHs selection by searching the energy center. It was assumed that nodes near CH also need significantly high energy. Therefore, CH needs to be in the center of energy. This technique exhibited a good result in improving energy consumption and increasing the network’s longevity.

Forming clusters efficiently contributes to the reduced energy consumption of WSNs. In [55], PSO was used for CH selection. Contrary to the traditional cluster-forming methods that only consider weight functions such as the distance between CH and non-CH nodes, the residual energy of CH, the distance from CH to BS, and the distance to CH are used to form clusters.

Unbalanced energy consumption and the premature death of nodes are some consequences of excessive transmission near CH. In [54], a new method was proposed to improve routing by conserving energy. The PSO algorithm was applied to search for the center of energy. The nodes closest to the energy center were considered as CH. The disadvantage of this method is the risk of data loss due to the uncertainty imposed by data collector movement. Conducting similar research, Vijayalakshmi and Anandan [49] relied on PSO to select CH with the optimal energy consumption in the cluster and enhance the capability of CH selection in WSN. The clusters were formed based on the distance of nodes to the base station (BS) and their energy level. Their proposed algorithm was effective at reducing the average loss rate of packets.

In [56], an optimization method was used to select the target nodes. As a result, the network longevity was extended by the selection of the target node minimization method. Additionally, a PSO algorithm was used to lower the transmission distance by enhancing the cluster nodes and optimizing the network’s energy consumption.

Another attempt for network energy consumption was made by Tam et al. [57], in which fuzzy clustering and a proposed PSO-based algorithm were used to reduce network interruptions. Both PSO and the genetic algorithm (GA) were employed to optimize the fuzzy C-means algorithm. The method was run repeatedly until an optimal sensor topology was achieved. The simulation results showed that the method improves the CH to BS connection rate as well as non-CH to CH connection rate and at the same time achieve reduced energy consumption. Naturally, having a longer connection duration in WSNs is desirable, which is why a PSO-based algorithm has been proposed to optimize the number of sensors that failed to connect to CH and the number of CHs that failed to connect to BS [57]. The objective function of the PSO was defined based on the total number of non-connected nodes in all clusters.

In [58], PSO was extended to dimensionality-based PSO (DPSO) [59] and hybrid dimensionality-based PSO (HDPSO) [60] for three dimensional localization of nodes. DPSO utilizes dimension-oriented optimization to locate the position of the target nodes. HDPSO uses a grouping method based on dimensional estimation to achieve fast convergence, which is favorable for intensive network deployment. The results indicated that their proposed methods exhibited better positioning accuracy and average time.

The demand for remote healthcare services based on IoT is rapidly growing. A natural choice for allocating cloud-server resources to users is using virtual machines (VM). Each VM is a virtual system using some portion of computational/storage capabilities of specific machine(s). Efficient VM selection is critical for providing high-quality remote services. In [61], an optimization algorithm for VM selection in IoT-based healthcare services was proposed using PSO, GA, and PPSO. The VM selection criteria were optimizing data storage time, rotation time, execution time, and medical-request waiting time. The reported simulation results indicate that the proposed model is better than the traditional model regarding data recovery efficiency and execution time.

During the data exchange on IoT, attackers may access the details of exchanged information. In [62], a multi-objective security-processing algorithm based on PSO was proposed that uses hierarchical clustering. In this algorithm, the goal was to hide secret information and to obtain the optimal particle. The basic concept is that a probability is assigned to each particle, leading to increased global diversity and restricting it from stopping at a local optimal.

In cloud computing, multi-objective PSO [63] can make resource selection efficient through the optimization of the middle layer residing between the service provider of the cloud and the client. The method tries to achieve the Pareto optimal solution to the cloud agent problem using multi-objective PSO. This algorithm could reduce not only response time but also energy consumption. However, the major drawback of the method is that increasing the number of iterations decreases the performance.

An attempt has been made to provide a concise and comprehensible understanding of machine intelligence in the field of biomedicine [64]. Furthermore, the ubiquitous nature of computational intelligence (CI) in biomedicine and emerging trends in CI-based healthcare and the IoMT are topics that have been addressed. A prime example of how CI is used in medical research is the prediction of breast cancer using PSO, which is discussed in this study. Finally, this research concludes with the perspectives on the biomedical industry based on computational intelligence.

Protecting patients’ sensitive and confidential data is one of the major challenges in the field of IoMT. In [65], an improved version of the PSO was proposed, and convergence and diversity have been enhanced using GA [66]. In real-world applications, complex optimization problems have more than one objective function. Therefore, this research presented a multi-objective version of the proposed algorithm. It has been used to optimize the key-based medical image encryption process to demonstrate its performance in real-world applications.

In another study devoted to medical image encryption, a creative cryptographic model with optimization strategies has been used [67]. Considering that patients’ data are often stored on a cloud server in the hospital, enforcing data security is critical. Therefore, a special framework is needed for the safe transmission and effective storage of medical images that are integrated with patient information. In this research, to increase the security level, the encryption and decryption processes were selected with the help of the optimal key using the combination of two swarm intelligence algorithms, i.e., grasshopper optimization [68] and PSO in elliptic curve encryption.

Data collection is an important process for the successful implementation of IoMT. Therefore, data collection and delivery must be carried out carefully. Missing data can affect the system’s overall performance and may be generated due to various factors such as bad connections, sensing errors, or external attacks. It is necessary to impute the missing data to avoid system-performance degradation. Once the data are received, they are divided into two groups: data without missing values and data with missing values. In [69], a dynamic adaptive network-based fuzzy inference system (ANFIS) was proposed to assign appropriate values to missing data. The proposed fuzzy system was trained using data with no missing values, while the data with missing values were used to impute them. The ANFIS was used in combination with PSO and GA. The final performance of the IoMT application was improved by 3% using ANFIS+PSO and by 5% using ANFIS+GA.

### 4.2. ACO in IoT/IoMT

There are multiple works on CH selection using the ACO algorithm. For example, a WSN-based multi-path routing using exponential ACO and fractional ABC was proposed in [70]. In the first step, considering several factors such as delay, distance, and energy, FABC was used to find CH. In the next step, EACO was used to discover the multi-path route. Compared to popular SI algorithms, EACO-FABC performance was superior in terms of efficiency and energy consumption. This method can also be adjusted to low-rate communication protocols with ease.

Optimal path planning directly impacts the amount of energy consumption in WSNs. Therefore, an improved version of ACO was used in [71] to find the optimal path for the mobile sink. The moving sink between CHs can be formulated as a traveling sales person (TSP) problem, and the ACO is applied to achieve an optimal route to cover all CHs. To minimize the energy consumption, the CH was rotated when its residual energy was lower than a predefined threshold. This optimization improves the longevity of the network and the delivery of data. The motivation for using a mobile sink is to avoid the so-called hot-spot problem in which the node near the static sink node will consume its energy much faster than other nodes. By moving the sink within the network, all network nodes will have approximately equal energy consumption, leading to a longer network lifetime.

In another attempt to deal with the hot spot problem in WSNs, the clustering of sensor nodes was done unequally [72]. In this approach, the size of the clusters closer to the master station (MS) was chosen to be smaller compared to the size of their counterparts farther away from MS. The multi-hop routing from CHs to MS was done using the ACO algorithm. Fuzzy logic was used to implement robust CH selection based on factors such as communication link quality, the number of neighbor nodes, and residual energy.

Scalability is a major design factor in WSN routing methods. Mobile sink and clustering were combined to propose a routing method based on ACO [71]. The performance was improved through heuristic factor optimization and finding the optimal motion path of the sink leading to extended longevity of the network. However, this method is not scalable, and its efficiency degrades as the number of nodes increases. In [73], an improved routing algorithm based on ACO was presented, which is more scalable and compatible with parameter changes in the network and accounts for the amount of consumed energy. The routing in IoT was formulated as a TSP-like problem, and ACO was utilized to solve it. The reported simulation results demonstrated the ability of the proposed method to reduce the amount of consumed energy and increase the nodes’ longevity. Large-scale WSNs may contain many sensor nodes distributed in an area [74]. In [75], a routing protocol in large-scale WSNs was proposed, keeping the information-transmission level and power-consumption level low. The protocol performs cluster division, albeit not based on CH. To avoid excessive energy consumption, the life of each ant was limited to its own community, i.e., the defined clusters. This algorithm can achieve a high delivery rate on a large-scale WSN.

Researchers have also tried to tackle the routing problem using hybrid SI approaches. For example, a multi-path routing WSN using modified versions of the ABC and ACO algorithms was proposed in [70]. The modified ABC was used to find the CH, followed by the application of the modified ACO for multi-path route discovery. Compared to other SI-based hybrid approaches, as the number of cycles increases, these modified SI methods outperform their standard counterparts in terms of energy consumption. However, by increasing the number of nodes, the advantage becomes negligible.

Sun et al. [76] proposed an enhanced routing algorithm relying on ACO while taking into account the communication transmission distance, the direction, and the residual energy of nodes. The authors also introduced a route-evaluation index to improve the pheromone update process. The proposed approach is better than other methods when the distance between the target and the sink node is long.

Enforcing security in WSNs data transmission is another important requirement. In [77], a secure routing protocol based on multi-objective ACO was proposed. The objective was to find better results for multi-path routing. This introduces the multi-objective optimization strategy of Pareto in ACO and uses the enhanced ACO to optimize the energy consumption of the nodes and routing problems. Compared to other security routing protocols that use ACO, this algorithm offers better performance regarding power consumption and packet loss rate.

For safety critical applications such as IoMT, WSN failure may lead to catastrophe. Therefore, the reliable deployment of WSN is highly desirable. The deployment problem has been tackled using ACO accompanied by a local search heuristic [78]. However, deploying nodes with minimum cost and constrained deployment reliability is NP-complete, so the authors have resorted to using ACO.

In [79], a systematic review focused on the most important algorithms used in the resource allocation process in IoMT has been conducted. ACO, DPSO, and basic PSO are the most important methods. It has been reported that DPSO is more efficient under heavy process congestion.

After horrifying car accidents, the time it takes to transfer the wounded to a hospital determines the survival or death of the wounded. ACO has been utilized to devise a method for choosing the optimal route for ambulances to reach the crash scene as fast as possible [80]. During the search for optimal routes, factors such as traffic and natural disasters are taken into account as well.

In the IoMT ecosystem, data sharing between different healthcare centers is quite natural and beneficial. With data sharing, transferring patients’ medical records from one center to another can be performed with ease. The shared data may be used for research purposes as well. Priyanka and Kaur [81] investigated remote data sharing between two hospitals in an IoMT setup. Finding the shortest path for data transfer is highly desirable, and it has been realized using ACO. Several parameters such as route length, route-detection time, and the efficient local threshold were considered in the proposed algorithms.

### 4.3. ABC in IoT/IoMT

As will be discussed in Section 5.1, for IoT/IoMT applications, the ABC method has been used less frequently than PSO and ACO, which is evident in the limited number of works reviewed in this section. Using the Sugeno fuzzy inference system, a centralized cluster-routing protocol was proposed [82]. The fuzzy rules were set with the aid of the ABC method. The proposed method outperformed existing fuzzy clustering algorithms in terms of lowering the intra-cluster gap, maximizing both the longevity of the network and the number of packets received by BS. The algorithm can also be extended to multi-hop routing and mobile sensor nodes.

In low-power and multi-functional WSNs, having access to a clustering mechanism with low energy consumption is vital. To this end, ABC has been modified based on an energy-efficient clustering protocol called BeeCluster [83]. The motivation behind the modification of ABC was the improvement of its exploitation capability.

QoS refers to using mechanisms or technologies in a network to ensure its performance via the reducing packet loss rate and transmission delay. A special type of mobile ad-hoc network for vehicular communication is VANET (vehicular ad hoc networks). QoS-restricted multicast routing has been found to be an NP-complete problem. The problem was cast to a search problem, and a modified version of the ABC method [73] was used to solve it [84]. Contrary to using a regular colony size, this method chooses a small colony to reduce the computational time in each cycle. The simulation results indicated that the method has the potential to achieve optimal performance.

Reducing system downtime in IoT applications is desirable, but it is a must in IoMT, which is a safety-critical application. This stems from the fact that the e-health system cannot be out of order for a long time. Otherwise, patients’ lives may be lost. A natural remedy to decrease the likelihood of system failure is the utilization of redundant components. In case one of the IoMT components fails, its the reserved counterpart will take its place immediately. With proper implementation, the transition between the failed component and the reserved one can be completed seamlessly. However, the budget devoted to redundant components is usually limited. That is why Santos et al. [85] have proposed an optimization model for maximizing an e-health system availability taking into account the limited budget for redundant components. The optimization model has been solved using ABC, differential evolution (DE), and GA.

ABC has also been used for diagnosing and classifying skin lesions using optimal segmentation and limited Boltzmann machines [86]. The proposed model includes a set of steps such as image acquisition, segmentation, feature extraction, and classification. Optimal classification is performed using ABC with Kapoor’s threshold. An accurate simulation analysis was carried out to evaluate the proposed model under various performance criteria.

It is necessary to monitor patients’ conditions to respond to their medical needs and manage their conditions, which can be realized using swarm optimization in IoMT [87]. In this method, clustering is done based on features and the distance between objects (i.e., devices) or groups. In the early stages, inspired by the BCO algorithm, data are collected and grouped. Some performance metrics are adopted to help minimize the required latency and computational costs. Various experiments with different parameter values for the proposed method were conducted to evaluate it against other clustering and optimization algorithms. Analyzing the results of experiments performed on different datasets such as Ward2ICU [88] revealed the superiority of the proposed approach.

### 4.4. Other SI Algorithms in IoT/IoMT

In [89], the proposed approach used the BFO algorithm to overcome the premature convergence of PSO. This algorithm optimizes the gap between cluster members and CH. It can enhance network longevity through the reduction in consumed energy. Using BOA, determining the optimal local node positions in WSN has been investigated as well [90]. Compared to PSO, BOA achieved more accurate node positioning and less computation time.

In IoMT-enabled healthcare systems, patients’ conditions can be monitored online, and their treatment can be done accordingly. This approach significantly helps to identify rare symptoms and conditions that may be present in patients based on the collected data. In addition, the advent of super-large-scale data generated from various sources has led to the emergence of big data, which has ignited a new competition among swarm intelligence algorithms. In [91], an optimization algorithm for big data analysis in IoMT using gravitational search optimization and the reflection-belief network with convolutional neural networks has been presented. In this algorithm, data optimization was performed on input data using gravitational search optimization. These data were collected to predict diabetes by estimating heart risk based on damage to heart arteries and nerves.

The performance of systems using big data is tightly bound to how efficient task scheduling is carried out. One possible approach is using Hadoop in a MapReduce framework. To this end, Senthilkumar [92] proposed a hybrid method combining firefly [93] and bat [94] algorithms for efficient energy-aware tasks scheduling. This hybrid method performs resource selection in the map phase and task scheduling in the reduce phase. A similar hybrid approach combines firefly and the genetic algorithm for energy-aware task scheduling at both the user and system levels in cloud-computing applications [95].

Data clustering is an inseparable part of big-data management. In [96], a comparative study was conducted on four SI methods, namely, bat, cuckoo [97], firefly, and PSO, and their time complexities were analyzed. These algorithms were used to cluster artificial and real medical datasets, and their effects on medical data mining were evaluated. Cuckoo clustering was reported as the slowest method. Firefly clustering was also slow in the presence of many agents, but PSO and bat algorithms were relatively faster than the other two algorithms. The measurements considered in the experiments were dimensions, the number of clusters, and the number of factors to select the best algorithm. It is clear that the optimized clustering process can improve IoT performance, and in this case, IoMT.

Another important component of big-data management is data classification. A new opposition-based learning BOA and multilayer perceptron was proposed for classifying medical data [98]. The model presented in this research operates in three stages: 1. preprocessing; 2. classification; and 3. parameter setting. An MLP is used as a classifier to determine the presence of diseases, in which the BOA is used to optimize the hyper-parameters of the MLP model. Using opposition-based learning helps increase BOA performance. The simulation analysis shows an improvement in the classification performance in the investigated dataset. However, more research is needed to improve the classification performance.

As discussed before, patients’ data security is very important in IoMT. In [99], a multi-watermarking approach for medical images was proposed to be used in the IoMT domain. The proposed method works based on the quantum random walk and brainstorming optimization algorithm [100]. During the multi-watermarking, patients’ private medical data were embedded in the medical images to achieve enhanced image security and ensure their authenticity. To validate the robustness, security, capacity, and intangibility of the multi-watermarking approach, a series of tests have been performed.

IoMT systems have to deal with a huge number of data that may be used for disease diagnosis, prediction, and monitoring. As storage and computational capacity is limited in some IoMT devices, patients’ medical data should be moved to cloud storage and external computing devices, respectively. This process can lead to security and privacy issues. To address these issues, a model based on swarm neural networks has been proposed to identify intruders in IoMT [101]. The proposed model seeks to identify intruders during data transmission as well as the possibility of analyzing healthcare data efficiently and accurately.

In [102], a bio-inspired scheme in IoMT setup assisted with blockchain, fog, and cloud computing has been proposed. This work seeks to minimize the cost of execution and the blocking of applications. Inspired by biology, robotics function blockchain task scheduling schemes were optimally assigned to existing nodes. The results of using this method in the field of IoMT show its superiority compared to other bio-inspired methods [103] regarding cost and data validation.

Another work devoted to CH selection in WSNs has been presented in [104]. The proposed method is called the augmented bio fold cuckoo search algorithm and aims to reduce overall energy consumption by wireless medical devices.

In [105], the authors try to develop an effective IoMT-based machine learning system for predicting the amount of health insurance, which includes three steps feature extraction, weight feature extraction, and forecasting. The prediction process uses a neural network (NN) to work with the weight feature vector, which is optimized by modified whale optimization (WOA) [106]. For a better comparison of the investigated papers, their important characteristics are summarized in Table 2.

## 5. Current and Future Trends

It is beneficial to investigate the number of conducted studies that have applied popular SI methods such as PSO, ACO, and ABC to address problems in the IoT/IoMT domain. Furthermore, such investigation sheds light on potential research directions for the future.

### 5.1. Current Trends on Using SI Methods in IoT/IoMT

Given the undeniable role of PSO, ACO, and ABC in IoT/IoMT applications, we devoted exclusive trend diagrams to them in Figure 7, but the statistics of other SI methods were summarized in a single trend diagram. The diagrams in Figure 7 were drawn according to Scopus’ search results from 1 January 2011, to 14 April 2022.

The most practical SI algorithm in the field of IoT/IoMT is PSO, which is used more than twice with respect to the other algorithms in this field. The popularity of PSO is due to its simplicity and fast convergence with fewer parameters compared to other algorithms, especially in solving continuous functions. PSO was originally inspired by how a group of birds maintains a specific formation during flight. Following a similar pattern, in PSO, the particles do not get too close to each other to avoid the collision, but they do not get too far away from each other either. A group of birds follows the same strategy and can also be used to organize multiple UAVs during flight. With the help of PSO, it is possible to position UAVs, plan their trajectory, and implement cooperation between them. This algorithm is also used for determining node location, routing, and CH selection.

ACO is the second-most-used SI method in IoT/IoMT. ACO was initially developed to model the path-finding behavior of ants. Given the history of ACO, it is a natural choice for path planning. Based on motivation similar to ACO, ABC has been proposed to capture collaboration among bees. It can be used to implement collaboration between multiple robots. ABC has nice features such as scalability, adaptive optimization, and distribution, so it is often utilized to address routing protocol problems. As can be seen in Figure 7, in terms of the number of published papers, SI algorithms other than PSO, ACO, and ABC have claimed third place, whereas ABC is in the last place.

### 5.2. SI Challenges in IoT/IoMT

Using SI algorithms in IoT/IoMT has many challenges. They can be divided into two categories. (1) The inherent challenges of SI algorithms, such as their high computational cost, the low efficiency of SI algorithms in local searches and simple search spaces, and the lack of guarantees in finding the optimal global are some of them. (2) On the other hand, there is a set of challenges in the field of IoMT that can be solved using SI algorithms. We have explained many of these challenges in this study, but more research is still needed. These challenges include efficient energy consumption; system modeling in IoMT; designing decision-support systems, data storage, and security; designing networks and communication systems; and supplying security for systems. Regarding challenges related to applying SI in IoT, the computational cost of SI methods can be dealt with using distributed computing paradigms such as MapReduce [107].

### 5.3. Future Trends

According to [70,108], the cognition of artificial bees is fully compatible with the WSN dynamic cognitive characteristics. Yet, such algorithms have been rarely used in WSN. Applying the ABC algorithm to wireless cognitive networks is one of the potential fields in which much research is needed.

Traditional QoS systems are not appropriate for WSNs, so more enhanced QoS frameworks for systems are expected. For instance, it is necessary to conduct more research on scalability and energy consumption efficiency for the physical layer, and the sleep mechanism for the sensor needs to be considered for energy saving. In addition, the network layer must balance the energy consumption. Despite all of the benefits of QoS, a limited amount of research has been performed on QoS quality. Therefore, researchers can potentially learn more about how to solve problems with QoS [109].

While edge computing reduces the response time of cloud computing by bringing some of the processing closer to the IoT users, the load balancing between the cloud server and edge devices still needs further investigation. SI methods have the potential to balance the load between cloud servers and edge devices [110].

### 5.4. Future Trends in IoMT

In order to provide high-quality medical care, human intelligence and high-quality technologies must be combined in an efficient and seamless manner. To this end, teams working in the field of IoMT should lead the process of device immunization and patient safety protection and improve connection or resource optimization [111]. The characteristics of SI algorithms clearly show that they have very good capabilities to address these challenges.

#### 5.4.1. Preserving Patient Safety

In any industry, security deficiency may lead to catastrophe, especially if human lives are involved. For example, a hospital cyberattack can affect critical equipment’s operation condition, endangering patients’ lives. IT teams working in the field of health information technology must take the necessary steps to neutralize cyberattacks in order to keep the patients’ safety. In developing necessary security mechanisms, it is crucial to consider the available resources and seek solutions to simplify security threats monitoring identification and neutralization [112].

#### 5.4.2. Supporting Connectivity in Critical Situations

The first step to keeping medical equipment connected to the network has a reliable infrastructure. This requires an agile, compatible, and secure network to support many devices and tools. In addition, the network should allow enforcement policy and prioritization rules to support equipment connectivity and ensure the precedence of critical equipment over ordinary equipment. Automating these rules across the network enables IT teams to identify possible network issues more effectively. This allows them to focus on strategic tasks related to improving safety and caring for patients [111].

#### 5.4.3. Dynamic Routing

It is quite likely that IoMT users have to carry monitoring devices such as wearables for long periods of time. Therefore, it is necessary to keep the wearable devices as light as possible, which puts a limit on the device battery capacity. Medical wearables are expected to stay functional for several hours or even days, which necessitates routing over WSNs with efficient energy consumption. In mobile wireless sensor networks (MWSNs) [113], the nodes are mobile-capable of changing network topology dynamically to lower transmission power between themselves. The population-based collaborative nature of SI methods can be utilized to implement the dynamic formation effectively.

## 6. Conclusions

As the global population and, in particular, the number of elderlies increase, meeting healthcare requirements becomes more challenging. IoMT is a promising aid to traditional healthcare systems, but it is still in its early stages, and much progress has to be made in IoT before the full potential of IoMT can be unlocked. In this paper, the potential and actual capabilities of SI algorithms and some of their applications in IoT/IoMT were investigated. Reliable connectivity technology with high quality is one of the primary prerequisites for the efficient implementation of IoT. The advantages of the 5G network, such as high capacity, high speed, low latency, and good service quality, have encouraged researchers to focus on the challenges of the 5G network. WSNs can play a vital role in implementing the information collection module of IoMT. However, due to the energy limitation of WSN nodes, there are key concerns in proper node finding and collaboration. Given that SI-based approaches can solve complicated NP problems and obtain reasonable solutions in WSNs, some of the solutions using SI methods for addressing concerns regarding use of WSNs were pointed out in this review.

Compared to analytical solutions, SI approaches are relatively easier to grasp and implement yet yield good solutions within a bearable amount of time. That is why (as reviewed in this paper) SI approaches have been used for various tasks related to IoMT, such as disease prediction, data security, missing value prediction, and resource allocation. Since IoMT end users do not necessarily stay at a fixed location (e.g., a hospital), portable sensors are critical for IoMT implementation. Wearable devices such as smartphones, smartwatches, etc., can act as portable sensors that users carry around in their daily activities. Inspired by the important role of wearables, they were briefly introduced in this review, and their market share was also presented. Despite being in its early stage, IoMT has already shown promising results. However, considering that medical diagnosis/treatment is a safety-critical domain, IoMT must be subject to much research before it can be entrusted with people’s lives. We tried to shed some light on the possible future research direction related to IoMT.

## Figures and Tables

**Figure 1 sensors-23-01466-f001:**
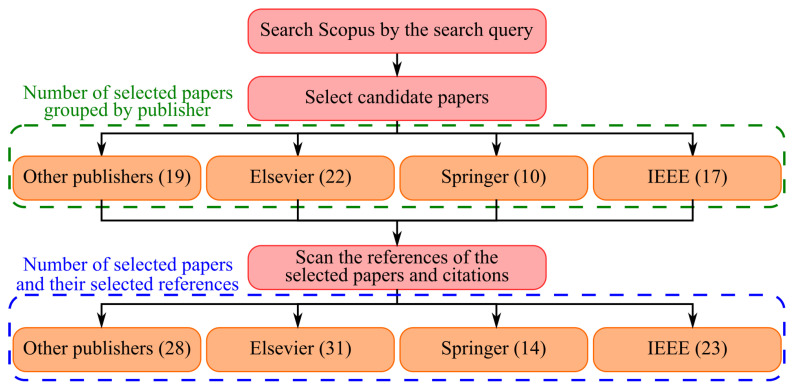
The number of reviewed papers.

**Figure 2 sensors-23-01466-f002:**
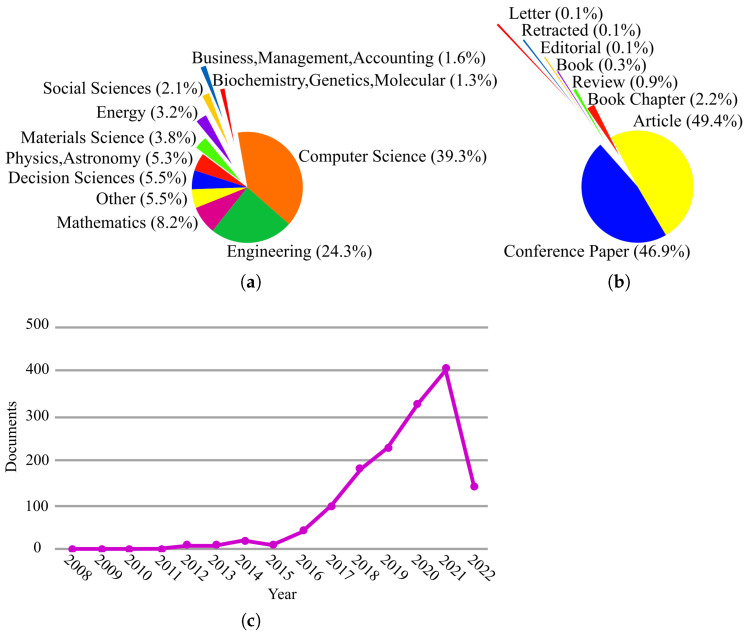
Search statistics extracted from Scopus related to the application of SI in IoT: (**a**) per subject area, (**b**) per publication type, and (**c**) per year.

**Figure 3 sensors-23-01466-f003:**
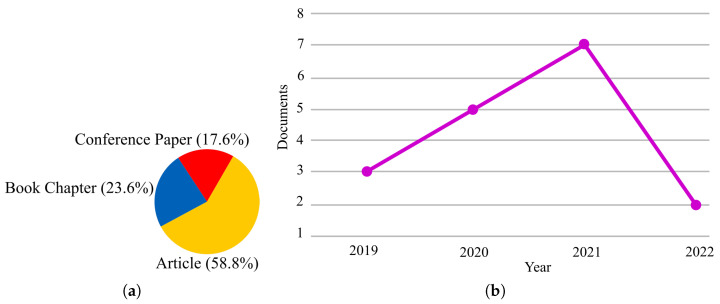
Search statistics extracted from Scopus related to the application of SI in IoMT: (**a**) per publication type, (**b**) per year.

**Figure 4 sensors-23-01466-f004:**
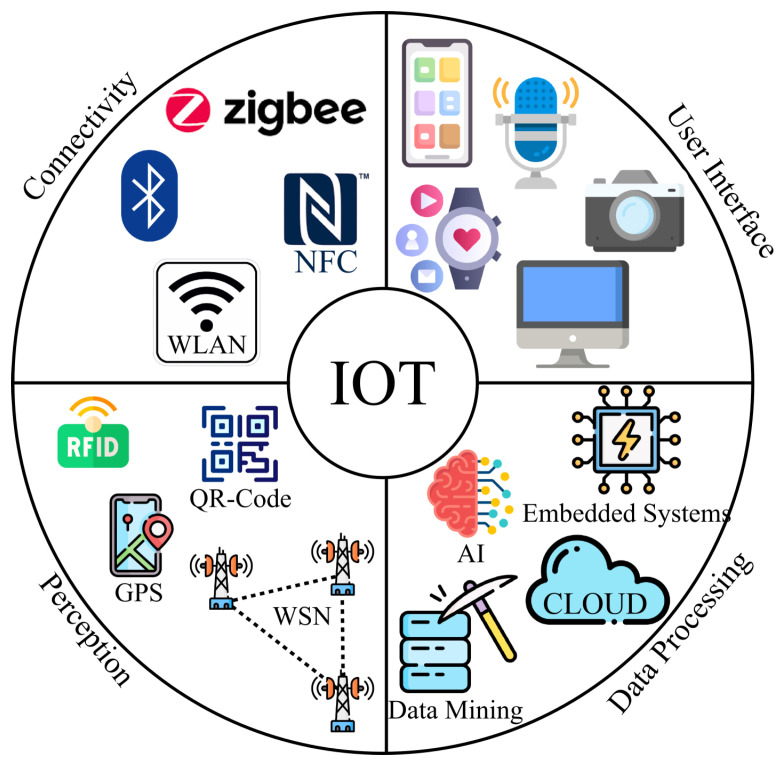
Components of IoT technology.

**Figure 5 sensors-23-01466-f005:**
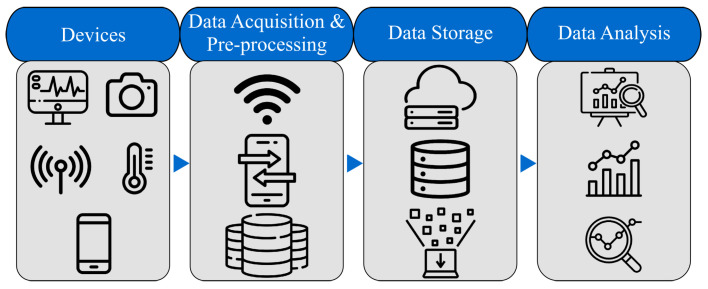
Stages of IoT in Healthcare (IoMT).

**Figure 6 sensors-23-01466-f006:**
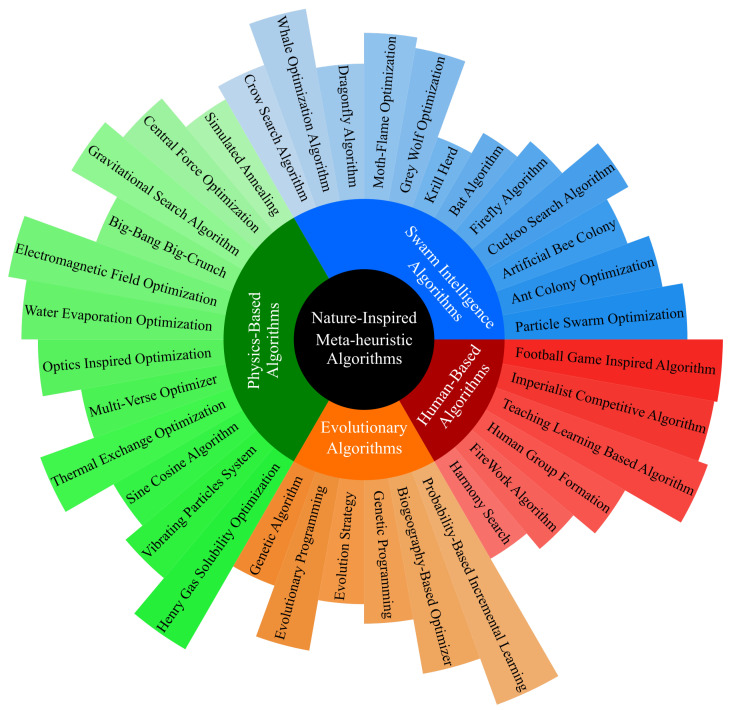
Nature-inspired meta-heuristic algorithms and the position of SI algorithms in them.

**Figure 7 sensors-23-01466-f007:**
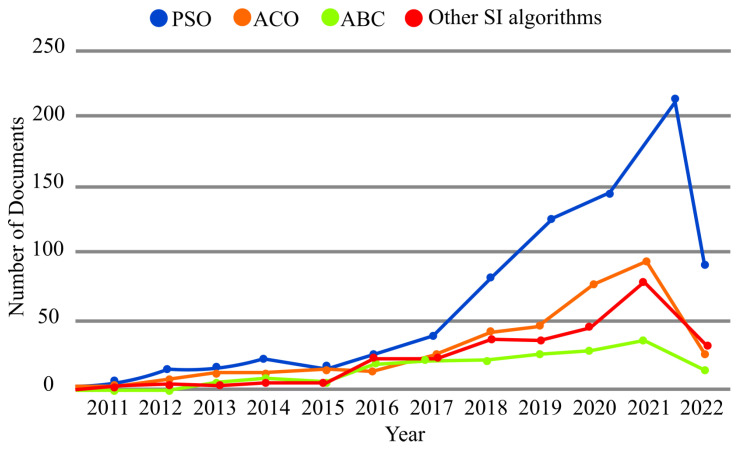
Statistics of PSO, ACO, ABC, and other SI algorithms in the field of IoT/IoMT.

**Table 1 sensors-23-01466-t001:** Comparison with previous related surveys.

Survey	Topics|Methods	Major Findings
[10], 2011	WSN Routing|SI	A recipe to define sound experiments and evaluation; anticipated growing application of SI in WSNs
[11], 2015	WSN|SI	Percentage of SI methods application for addressing WSNs different issues
[16], 2020	SIoT|DL	Importance of SIoT in various applications, especially diagnosis methods for heart diseases and brain tumors; high accuracy is achievable in disease diagnosis due to big-data methods used in SIoT
[7], 2020	IoT|SI	Important challenges in WSN applications and SI methods capabilities to deal with them; analyzed three categories of the wireless network aided with UAV and role of SI in their applications; combining UAV with 5G, IoT, etc. is a worthy direction for future works
[17], 2021	IoMT|ML	Without special considerations, e.g., resource and time complexity, etc., traditional ML fails to address IoMT security and privacy issues; the majority of reviewed studies ignore these considerations; and future ML-based approaches should comply with these considerations.
[18], 2022	IoMT|ML, DL	Investigated different ML/DL-based attack detection techniques; future works on enforcing IoMT security using defensive techniques.
[19], 2022	IoMT|ML	Investigated ML-based intrusion detection for IoMT; presented requirements and threats affecting IoMT security; and presented advantages and disadvantages of ML-based solutions and their ability for IoMT security
[20], 2021	IoMT|DL	Reviewed characteristics and challenges of IoMT for the multimedia event processing based on IoT service-oriented architecture (SoA) of IoT; current approaches are robust but not adaptable due to their user interface shortcomings and limited vocabulary; challenges of using DL for multimedia event processing; and the inability of existing object detection approaches for achieving a minimum time of response while keeping the accuracy high
[12], 2018	IoT|SI	Highly popular SI methods in IoT: ACO, PSO, and ABC
Our paper, 2022	IoT, IoMT|SI	Investigation of SI algorithms used in the IoT with a special emphasis on their application for improving IoMT technology

**Table 2 sensors-23-01466-t002:** Summary of analyzed works.

Ref.	System|Method	Goal
[49], 2019	IoT|PSO	Optimizing WSN routing
[50], 2017	IoT|PSO	Optimizing WSN routing
[51], 2017	IoT|PSO	Finding fault-tolerant routing methods
[52], 2018	IoMT|PSO	Covering a specific area with a network of sensors
[54], 2019	IoT|PSO	Preventing energy-hole generation and performing CHs selection
[56], 2016	IoT|PSO	Lowering the transmission distance by enhancing the nodes in the cluster and optimizing the energy consumption of the network
[57], 2018	IoT|PSO+GA	Reducing interruptions in networks
[58], 2018	IoT|PSO	Locating the position of the target nodes
[61], 2018	IoT|PSO+GA	VM selection
[62], 2019	IoT|PSO	Hiding secret information
[63], 2016	IoT|PSO	Optimizing resource selection
[64], 2022	IoMT|PSO	Breast cancer prediction
[65], 2021	IoMT|PSO+GA	Protecting patients’ sensitive and confidential data
[67], 2020	IoMT| Grasshopper Optimization + PSO	Encrypting medical image
[69], 2019	IoMT| ANFIS + PSO + GA	Assigning appropriate values to missing data
[70], 2017	IoT|ACO+ABC	A WSN-based multi-path routing and CH selection
[71], 2018	IoT|ACO	Finding an optimal path for mobile sink
[72], 2016	IoT|ACO+Fuzzy Logic	Dealing with the hot-spot problem in WSNs
[73], 2017	IoT|ACO	Reducing the amount of energy consumed, increasing node longevity, and improving routing algorithm
[75], 2017	IoT|ACO	Keeping information transmission-level high and power consumption level low
[76], 2017	IoT|ACO	Proposing an enhanced routing algorithm
[77], 2019	IoT|ACO	Enforcing security in WSNs data transmission and finding better results for multi-path routing
[78], 2017	IoT|ACO	Reliable deployment of WSN
[80], 2018	IoMT|ACO	Choosing the optimal route for ambulances to minimize the time to reach the crash scene
[81], 2018	IoMT|ACO	Finding the shortest path for data transfer
[82], 2017	IoT|ABC	Proposing centralized cluster routing protocol
[83], 2019	IoT|ABC	Clustering with low energy consumption in WSNs
[84], 2017	IoT|ABC	Reducing the computational time
[85], 2022	IoMT| ABC + DE + GA	Maximizing e-health system availability, taking into account the limited budget for redundant components
[86], 2021	IoMT|ABC	Skin-lesion diagnosis and classification
[87], 2021	IoMT|ABC	Offering solutions for patients’ data analysis and management; clustering data (patients)
[89], 2017	IoT|BFO	Optimizing the gap between cluster members and CH and enhancing network longevity through the reduction in consumed energy
[90], 2017	IoT|BOA	Finding the local node optimal position
[91], 2022	IoMT|gravitational search optimization + reflection-belief networks + CNN	Optimizing data to predict diabetes
[96], 2019	IoMT|Bat + cuckoo+ firefly + PSO	Clustering artificial and real medical datasets
[98], 2021	IoMT|BOA + multilayer perceptron	medical data classification
[99], 2022	IoMT|quantum random walk + brainstorming optimization	Embedding private medical data in private images to achieve enhanced image security and ensure their authenticity
[101], 2021	IoMT|swarm NNs	Identifying intruders in the IoMT data-driven system and finding a solution to identify intruders during data transmission as well as the possibility of analyzing healthcare data efficiently and accurately
[102], 2021	IoMT|bio-inspired method	Minimizing the cost of execution and blocking of applications
[104], 2021	IoMT|augmented bio fold cuckoo	Selecting CH in WSNs, reducing overall energy consumption by wireless medical devices
[105], 2020	IoMT|WOA + NNs	Predicting the amount of health insurance

## Data Availability

The data that support the findings of this study are available on request from the corresponding author.

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
