# Peer review of "Swarm Intelligence in Internet of Medical Things: A Review"

_sensors, 2023, doi:10.3390/s23031466_

Round 1
Reviewer 1 Report
Congratulation for your work on preparing the paper. Comments about the paper are presented next.
The paragraph between lines 37-48 should be rewritten, the ideas are not presented clearly.
The search strategy seems childish - the information provided in lines 50-64 cand be completely deleted.
In Fig 2, a) and b), the font size is too small, the subject areas and the type of the document cannot be read.
Chaper 2 IoT overview - there is no new information presented in this chapter, thus can be elimitated.
More detailed Future trends chapters (5.3 and 5.4) will add more value to the paper. Basicaly, there are only four directions specified in the paper – two directions in IoT (ABC algorithm in wireless cognitive networks, QoS research), and two directions in IoTM (security mechanisms, automating connectivity of medical devices).
Author Response
Congratulation for your work on preparing the paper. Comments about the paper are presented next.
1-The paragraph between lines 37-48 should be rewritten, the ideas are not presented clearly.
Ans: The aforementioned paragraph has been revised as below:
Using IoT in medical domain leads to internet of medical things (IoMT). In this article, the key researches related to applications of SI algorithms in the IoT domain with special focus on IoMT are reviewed. Given that WSN is one of the primary technologies for IoT realization, the SI applications in four major subcategories of WSN i.e. node localization, sensor deployment, routing protocol, and CH selection are reviewed. Appropriate coverage of the environment is an important factor in WSNs deployment. However, the number of static nodes is limited and may not be enough to cover the whole area appropriately. Existing literature on dynamic coverage of the environment using swarm of unmanned aerial vehicles (UAVs) [16] is reviewed as well. Apart from dynamic coverage, other issues related to WSNs such as energy limitation are also reviewed.
The ultimate goal of IoMT is providing better medical service to patients. One of the critical requirements of IoMT is that it must be affordable and available anywhere and anytime. Such requirements can be realized by exploiting portable devices such as wearables that IoMT users already own. The applications of wearables in IoMT are reviewed as well. The article is ended with closing thoughts on possible future directions for further development of IoMT.
2-The search strategy seems childish - the information provided in lines 50-64 can be completely deleted.
Ans: The mentioned lines were deleted.
3-In Fig 2, a) and b), the font size is too small, the subject areas and the type of the document cannot be read.
Ans: Fig 2.a and 2.b have been revised for better font clarity.
4-Chaper 2 IoT overview - there is no new information presented in this chapter, thus can be elimitated.
Ans: Removing the aforementioned chapter altogether hurts the integrity of the article. Therefore, it has been shortened as below:
The IoT consists of four primary components which are perception, connectivity, data processing, and user interface as shown in Figure 4. In the perception component, various sensors and devices such as GPS, QR-Code, RFID, etc., are used to gather vital information from the environment in which IoT operates. The transfer of collected data to users and/or cloud servers for further processing and storage is performed using communication technologies such as Bluetooth, Zigbee, WLAN, etc. The collected data are processed using combination of cloud and edge computing. The IoT interface with users is realized via different devices such as smartphones, smartwatches, microphones, cameras, monitors, etc. [24].
5-More detailed Future trends chapters (5.3 and 5.4) will add more value to the paper. Basicaly, there are only four directions specified in the paper – two directions in IoT (ABC algorithm in wireless cognitive networks, QoS research), and two directions in IoTM (security mechanisms, automating connectivity of medical devices).
Ans: We added more content regarding future research. The added parts are repeated below:
… Regarding challenges related to applying SI in IoT, the computational cost of SI methods can be dealt with using distributed computing paradigms such as MapReduce [107].
5.3. Future Trends
… While edge computing reduces response time of cloud computing by bringing some of the processing closer to the IoT users, the load balancing between cloud servers and edge devices still needs further investigation. SI methods have the potential to balance the load between cloud servers and edge devices [110].
5.4.3. Dynamic routing
It is quite likely that IoMT users have to carry monitoring devices such as wearables for long periods of time. Therefore, it is necessary to keep the wearable devices as light as possible which puts limit on the device battery capacity. Medical wearables are expected to stay functional for several hours or even days which demands routing over WSNs with efficient energy consumption. In Mobile wireless sensor networks (MWSNs) [113], the nodes are mobile capable of changing network topology dynamically to lower transmission power between themselves. The population-based collaborative nature of SI methods can be utilized to implement the dynamic formation effectively.

Reviewer 2 Report
1. Kindly proofread the entire manuscript once.
2. To begin with, there are some typos and grammar mistakes. Some long sentences could make readers confused.
3. Check Figure 1 title, too lengthy.
4. Figure 2 and 3 and the subfigures quality need to be improved.
5. Use all figures in HD and standard size and quality.
6. Check the alignment of all tables, for example, table 1 alignment.
7. Check figures 5,6 and 7 and redraw
8. Check Table 2. Summary of analyzed works
9. Last column, 1: Grasshopper Optimization 2: Fuzzy Logic alignment issue??
10. Check all figures and tables, alignment, and captions once.
11. Check all references, a few items missing and not in the correct format.
12. Could the authors clarify why they excluded documents extracted (e.g., book chapter, conference review, and review paper in the revised paper) for this study?
13. How could/should your study help future studies?
refer :
10.1504/IJAIP.2020.107008
10.2478/cait-2018-0031
Author Response
- Kindly proofread the entire manuscript once.
Ans: The paper has been proofread.
- To begin with, there are some typos and grammar mistakes. Some long sentences could make readers confused.
Ans: We did our best to fix the typos in the paper. The modified parts have been marked with blue color.
- Check Figure 1 title, too lengthy.
Ans: The caption of Figure 1 has been revised. Moreover, Figure 1 has been redrawn for better readability.
- Figure 2 and 3 and the subfigures quality need to be improved.
Ans: Figures 2 and 3 have been redrawn for better clarity.
- Use all figures in HD and standard size and quality.
Ans: Figures 1 to 7 have been redrawn to improve their quality.
- Check the alignment of all tables, for example, table 1 alignment.
Ans: Alignment of the tables has been fixed.
- Check figures 5,6 and 7 and redraw
Ans: All figures have been redrawn for better readability.
- Check Table 2. Summary of analyzed works. Last column, 1: Grasshopper Optimization 2: Fuzzy Logic alignment issue??
Ans: The last row of the table was devoted to some abbreviations i.e. GO and FL that were used in the table. In the revised version, we removed the footnotes and used the full form of GO and FL which are Grasshopper Optimization and Fuzzy Logic.
- Check all figures and tables, alignment, and captions once.
Ans: As mentioned in the answers to the previous comments, we have redrawn the figures for better clarity and fixed alignment of the tables. The long caption of figure 1 has been shortened as well.
- Check all references, a few items missing and not in the correct format.
Ans: The following references were revised:
- Publisher city (Berlin) was added to reference: “Complex Social and Behavioral Systems: Game Theory and Agent-based Models”
- Kakhi, K.; Alizadehsani, R.; Kabir, H.D.; Khosravi, A.; Nahavandi, S.; Acharya, U.R. The internet of medical things and artificial intelligence: trends, challenges, and opportunities. Biocybernetics and Biomedical Engineering 2022, 42, 749–771
- Wang, J.; Gao, Y.; Liu, W.; Sangaiah, A.K.; Kim, H.J. An improved routing schema with special clustering using PSO algorithm for heterogeneous wireless sensor network. Sensors 2019, 19, 671–687
- Alrazgan, M. Internet of medical things and edge computing for improving healthcare in smart cities. Mathematical Problems in Engineering 2022, 2022, 1–10
- Santos, G.L.; Gomes, D.; Silva, F.A.; Endo, P.T.; Lynn, T. Maximising the availability of an internet of medical things system using surrogate models and nature-inspired approaches. International Journal of Grid and Utility Computing 2022, 13, 291–308.
- Sampathkumar, A.; Tesfayohani, M.; Shandilya, S.K.; Goyal, S.; Shaukat Jamal, S.; Shukla, P.K.; Bedi, P.; Albeedan, M. Internet of Medical Things (IoMT) and Reflective Belief Design-Based Big Data Analytics with Convolution Neural Network-Metaheuristic Optimization Procedure (CNN-MOP). Computational Intelligence and Neuroscience 2022, 2022, 1–14.
- Yan, F.; Huang, H.; Yu, X. A multi-watermarking scheme for verifying medical image integrity and authenticity in the Internet of Medical Things. IEEE Transactions on Industrial Informatics 2022, 18, 8885–8894.
- Mohammed, M.A.; Ibrahim, D.A.; Abdulkareem, K.H.; et al. Bio-inspired robotics enabled schemes in blockchain-fog-cloud assisted IoMT environment. Journal of King Saud University-Computer and Information Sciences 2021. https://doi.org/10.1016/j.jksuci.2021.11.009
- Hamdi, M.; Zaied, M. Resource allocation based on hybrid genetic algorithm and particle swarm optimization for D2D multicast communications. Applied Soft Computing 2019, 83, 105605–105618.
- Lewandowski, M.; PÅ‚aczek, B. An event-aware cluster-head rotation algorithm for extending lifetime of wireless sensor network with smart nodes. Sensors 2019, 19, 4060–4080.
- Could the authors clarify why they excluded documents extracted (e.g., book chapter, conference review, and review paper in the revised paper) for this study?
Ans: As was mentioned in the last sentence of the first paragraph of the Search Strategy Section: The paper selection was done based on criteria such as paper publisher, citation count, and relevance to IoT/IoMT/SI. So, we did not exclude documents extracted because they are book chapters, conference reviews, and review papers. In Table 1, we even clarify the features of previous review papers and our paper.
- How could/should your study help future studies?
Ans: We have pointed out challenges of IoT/IoMT and how the potential future works for addressing these challenges using Swarm Intelligence methods.
- Refer :
10.1504/IJAIP.2020.107008
10.2478/cait-2018-0031
Ans: The papers mentioned above have been referred to in the paper. The related part is repeated below for ease of reference:
“The performance of systems using big data is tightly bound to how efficient task scheduling is done. One possible approach is using Hadoop in a MapReduce framework. To this end, Senthilkumar [92] proposed a hybrid method combining firefly [93] and bat [94] algorithms for efficient energy-aware tasks scheduling. This hybrid method performs resource selection in the Map phase and task scheduling in the Reduce phase. A similar hybrid approach combines firefly and genetic algorithm for energy-aware task scheduling at both user and system levels in cloud computing applications [95].”
